# Dimensionless Correlations for Natural Convection Heat Transfer from a Pair of Vertical Staggered Plates Suspended in Free Air

Alessandro Quintino [1,*], Marta Cianfrini [2], Ivano Petracci [3], Vincenzo Andrea Spena [1] and Massimo Corcione [1]

[1]   Dipartimento di Ingegneria Astronautica Elettrica ed Energetica, Sapienza Università di Roma,
     Via Eudossiana 18, 00184 Roma, Italy; vincenzo.spena@uniroma1.it (V.A.S.);
     massimo.corcione@uniroma1.it (M.C.)
[2]   Istituto Nazionale di Fisica Nucleare—Laboratori Nazionali di Frascati, Divisione Tecnica,
     Via Enrico Fermi 54, 00044 Frascati, Italy; marta.cianfrini@lnf.infn.it
[3]   Dipartimento di Ingegneria Industriale, Università di Roma Tor Vergata, Via del Politecnico 1,
     00133 Roma, Italy; ivano.petracci@uniroma2.it
*   Correspondence: alessandro.quintino@uniroma1.it

**Abstract:** Buoyancy-induced convection from a pair of staggered heated vertical plates suspended in free air is studied numerically with the main scope to investigate the basic heat and momentum transfer features and to determine in what measure any independent variable affects the thermal performance of each plate and both plates. A computational code based on the SIMPLE-C algorithm for pressure-velocity coupling is used to solve the system of the governing conservation equations of mass, momentum and energy. Numerical simulations are carried out for different values of the Rayleigh number based on the plate length, as well as of the horizontal separation distance between the plates and their vertical alignment, which are both normalized by the plate length. It is observed that an optimal separation distance between the plates for the maximum heat transfer rate related to the Rayleigh number and the vertical alignment of the plates does exist. Based on the results obtained, suitable dimensionless heat transfer correlations are developed for each plate and for the entire system.

**Keywords:** natural convection in free air; vertical staggered plates; buoyancy-induced convection; dimensionless correlations



## 1. Introduction

Buoyancy-induced convection in air from heated vertical parallel plates is of much interest for a number of thermal engineering applications, such as the electronic equipment cooling and the solar energy capture to name a few.

The first documented work dealing with this subject was executed experimentally by Elenbaas [1], who, by using a pair of square plates suspended face to face in free air, obtained the optimal plate spacing for the dissipation of the maximum amount of heat. Investigations aimed at determining the optimal plate spacing for vertical parallel plate channels subjected to uniform wall temperature and uniform heat flux conditions were subsequently conducted theoretically by Bodoia and Osterle [2], Levy [3], Anand et al. [4] and Bar-Cohen and Rohsenow [5], as well as both experimentally and theoretically by Onur et al. [6,7] and Baskaya et al. [8] for channels consisting of a heated plate and an unheated plate insulated at the rear.

Other studies dealing with natural convection in vertical parallel plate channels, which were mostly performed experimentally, were carried out by Aung et al. [9], Carpenter et al. [10], Sparrow and Bahrami [11], Wirtz and Stutzman [12], Azevedo and Sparrow [13], Webb and Hill [14], Martin et al. [15], Straatman et al. [16], Qing et al. [17] and

Lewandowski et al. [18] for both conditions of symmetric and asymmetric heating, with the aim to obtain heat transfer correlations rather than to determine the optimal plate spacing.

While in some engineering applications the plates could be staggered rather than positioned face to face, only two studies are readily available in the literature on buoyancy-driven convection from a pair of staggered vertical parallel plates. One was performed numerically by Acharya and Jang [19] and the other was executed experimentally by Tanda [20]. Their common conclusion was that for any assigned plate spacing, an optimal plate staggering for maximum overall heat transfer rate does exist. In particular, they found that the staggering of the plates resulted in a considerable enhancement of the thermal performance of the lower plate and a relatively moderate degradation of the thermal performance of the upper plate. However, no correlation was developed and this was mainly due to the limited number of investigated configurations. Other works with a bearing on the subject are those executed for arrays of fully staggered plates, such as those authored by Sparrow and Prakash [21,22], Guglielmini et al. [23], Tanda [24] and Ledezma and Bejan [25].

Framed in this general background, which points out a meaningful lack of data, a comprehensive investigation of natural convection from pairs of staggered vertical plates heated at the same uniform temperature and suspended in free air is performed numerically. The study is conducted using the Rayleigh number based on the plate length, the horizontal plate spacing normalized by the plate length and the vertical plate staggering normalized by the plate length as controlling parameters. The main scope of the paper is to scrutinize the basic heat and momentum transfer features and to analyze how the independent variables affects the thermal performance of each plate and the pair of plates. Furthermore, the study also aims to develop dimensionless heat transfer correlating equations that will be useful for thermal engineering applications.

## 2. Materials and Methods

### 2.1. Mathematical Formulation

A pair of thin staggered vertical plates of length $H$ and infinite width are suspended in free air at a distance $W$ to form a duct, as sketched in Figure 1, where the reference Cartesian coordinate system is also indicated. The edges of the plates are misaligned so that $L$ is the portion of one plate facing the other, which means that when $L = H$ the plates are located face to face and when $L = 0$ the plates are completely staggered. The front side of each plate is heated at a uniform temperature $t_h$ and the surrounding undisturbed fluid reservoir is maintained at a lower uniform temperature $t_\infty$. The rear side of each plate is assumed to be perfectly insulated. The resulting buoyancy-induced flow is considered to be two-dimensional, laminar and incompressible, with constant physical properties. Additionally, the buoyancy effects on the momentum transfer are taken into account through the customary Boussinesq approximation, whereas viscous dissipation and pressure work are neglected.

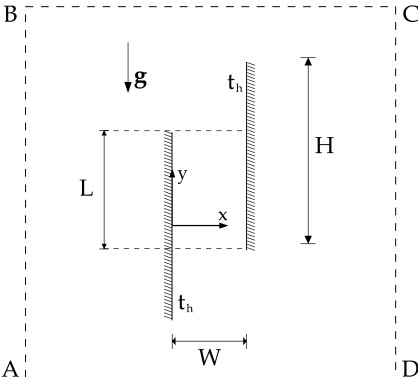

**Figure 1.** Sketch of the geometry, coordinate system and computational domain.

Upon incorporating these hypotheses into the equations of continuity, momentum and energy, the following set of governing equations expressed in dimensionless form is obtained:

$$\nabla \cdot \mathbf{V} = 0 \tag{1}$$

$$\frac{\partial \mathbf{V}}{\partial \tau} + (\mathbf{V} \cdot \nabla)\mathbf{V} = -\nabla P + \nabla^2 \mathbf{V} - \frac{Ra}{Pr} T \frac{\mathbf{g}}{g} \tag{2}$$

$$\frac{\partial T}{\partial \tau} + (\mathbf{V} \cdot \nabla)T = \frac{1}{Pr} \nabla^2 T \tag{3}$$

where $\tau$ is the dimensionless time normalized by $H^2/\nu$, $\mathbf{V}$ is the dimensionless velocity vector possessing horizontal and vertical components $U$ and $V$ normalized by $\nu/H$, $T$ is the dimensionless temperature excess over the uniform temperature of the undisturbed fluid reservoir normalized by the temperature difference $(t_h - t_\infty)$, $P$ is the dimensionless sum of the thermodynamic and hydrostatic pressures normalized by $\rho\nu^2/H^2$, $\mathbf{g}$ is the gravity vector, $Pr = \nu/\alpha$ is the Prandtl number and $Ra_H$ is the Rayleigh number defined as the following:

$$Ra_{\mathrm{H}} = \frac{g\beta(t_h - t_\infty)H^3}{\alpha\nu} \tag{4}$$

in which $\nu$ is the kinematic viscosity, $\rho$ is the mass density, $\alpha$ is the thermal diffusivity and $\beta$ is the coefficient of volumetric thermal expansion of the fluid.

The related boundary conditions are $\mathbf{V} = 0$ and $T = 1$ at the heated side of any plate and $\mathbf{V} = 0$ and $\partial T/\partial n = 0$ at the thermally insulated rear side of any plate. $n$ denotes the normal direction relative to the plate surface and $\mathbf{V} = 0$ and $T = 0$ at very large distance from the plates.

The integration domain is taken as a two-dimensional square ABCD containing the plates and extending sufficiently far from them. Such an integration domain is filled with a non-uniform Cartesian grid possessing a higher concentration of grid lines near the plates and a lower concentration of grid lines far away from the plates. Notice that due to the infinitesimal thickness assumed for the plates, each grid node situated on any plate acutally consists of a pair of coincident yet distinct grid nodes. One belongs to one side of the plate and the other belongs to the opposite side of the plate such that the same spatial location is permitted to possess simultaneously two different temperatures, with one for each side of the plate.

The boundary conditions required for the numerical solution of the governing Equations (1)–(3) have to be specified at each of the boundary lines that enclose the two-dimensional integration domain defined above. Once these lines are set sufficiently far away from the plates, the motion of the fluid which enters or leaves the computational domain can be reasonably assumed to normally occur relative to them. The entering fluid is assumed to be at the undisturbed free stream temperature. In contrast, due to the fact that the temperature of the outgoing fluid is not known a priori, a zero temperature gradient along the normal to the boundary line is assumed and thus implies that the local heat transfer is dominated by convection rather than by conduction, provided that the outflow velocity is sufficiently large.

Accordingly, the following boundary conditions are applied:

(a) At the heated side of any plate, $U = 0$, $V = 0$ and $T = 1$;
(b) At the thermally insulated rear side of any plate, $U = 0$, $V = 0$ and $\partial T/\partial X = 0$;
(c) At the left boundary line A–B, $\partial U/\partial X = 0$, $V = 0$ and $T = 0$ if $U > 0$ or $\partial T/\partial X = 0$ if $U < 0$;
(d) At the top boundary line B–C, $U = 0$, $\partial V/\partial Y = 0$ and $T = 0$ if $V < 0$ or $\partial T/\partial Y = 0$ if $V > 0$;
(e) At the right boundary line C–D, $\partial U/\partial X = 0$, $V = 0$ and $T = 0$ if $U < 0$ or $\partial T/\partial X = 0$ if $U > 0$;

(f)     At the bottom boundary line A–D, $U = 0$, $\partial V/\partial Y = 0$ and $T = 0$ if $V > 0$ or $\partial T/\partial Y = 0$ if $V < 0$;

in which $X$ and $Y$ stand for the dimensionless Cartesian coordinates normalized by $H$.

The initial conditions assumed throughout the integration domain are fluid at rest, i.e., $U = V = 0$, and uniform temperature $T = 0$.

### 2.2. Computational Procedure

The system of the governing equations defined by Equations (1)–(3), in combination with the boundary and initial conditions stated earlier, is solved through a control-volume formulation of the finite-difference method using an in-house developed computer code. The pressure-velocity coupling is handled using the SIMPLE-C algorithm introduced by Van Doormaal and Raithby [26], which is essentially a more implicit variant of the SIMPLE algorithm developed by Patankar and Spalding [27]. The convective terms are approximated through the QUICK discretization scheme proposed by Leonard [28], whereas a second-order backward scheme is applied for time integration. Time discretization is chosen uniformly.

Starting from the assigned initial fields of the dependent variables, at each time-step the system of the discretized algebraic governing equations is solved iteratively by the method of a line-by-line application of the Thomas algorithm. A standard under-relaxation technique is enforced in all steps of the computational procedure to ensure an adequate convergence. Within each time-step, the spatial numerical solution of the velocity and temperature fields is considered to be converged when the maximum absolute value of the mass source and the relative changes of the dependent variables at any grid-node between two consecutive iterations are smaller than the pre-specified values of $10^{-6}$ and $10^{-7}$, respectively. Time-integration is stopped once steady state is reached. This means that the simulation procedure ends when the relative changes of the time-derivatives of the dependent variables at any grid-node between two consecutive time-steps is smaller than the pre-set value of $10^{-8}$.

After convergence of the velocity and temperature fields is satisfactorily attained, the pair of average Nusselt numbers for the lower plate, $Nu_{\mathrm{L}}$, and for the upper plate, $Nu_{\mathrm{U}}$, are calculated as follows:

$$Nu_{\mathrm{L}} = \int_{-\frac{H}{2}}^{+\frac{H}{2}} -\frac{\partial T}{\partial X}\bigg|_{h} dY \tag{5}$$

$$Nu_{\mathrm{U}} = \int_{-\frac{H}{2}}^{+\frac{H}{2}} -\frac{\partial T}{\partial X}\bigg|_{h} dY \tag{6}$$

where subscript $h$ denotes the heated side of each plate. The temperature gradient is evaluated by a second-order temperature profile embracing the wall-node and the two adjacent fluid-nodes, whereas the integral is calculated numerically by means of the trapezoidal rule. Additionally, the average Nusselt number of the whole system, which is identified as $Nu$, is calculated as the arithmetic mean of the average Nusselt numbers of both plates and is defined as follows.

$$Nu = \frac{Nu_{\mathrm{L}} + Nu_{\mathrm{U}}}{2} \tag{7}$$

Numerical tests on the dependence of the results obtained on the mesh spacing and time stepping, as well as on the extent of the computational domain, have been methodically performed for several combinations of the independent variables, namely $Ra_{\mathrm{H}}$, $W/H$ and $L/H$. Accordingly, the grid-spacings and time-steps used for computations in combination with assigned extents of the integration domain are chosen in such a manner that further grid and time stepping refinements or extensions of the integration domain do not produce noticeable modifications either in the flow field or in the heat transfer rates, with percentage

changes smaller than the pre-established accuracy value of 1%. The typical number of nodal points used for simulations lies in the ranges between $100 \times 100$ and $200 \times 200$, whilst the horizontal and vertical extents of the whole integration domain span between 5 and 10 times the plates length $H$. Moreover, typical dimensionless time-steps used for simulations lie in the range between $10^{-4}$ and $10^{-5}$. Selected results of the grid-size and time-stepping sensitivity analysis are presented in Tables 1 and 2.

**Table 1.** Grid sensitivity analysis for $W/H = 0.5$, $L/H = 0$ and 1, $Ra_\mathrm{H} = 10^4$ and $10^5$.

| $W/H$ | $L/H$ | $Ra_\mathrm{H}$ | Mesh Size | $Nu_\mathrm{L}$ | % | $Nu_\mathrm{U}$ | % |
|-------|-------|---------|-----------|------|------|------|------|
| 0.5 | 1 | $10^4$ | $80 \times 80$ | 6.18 | – | 6.18 | – |
| | | | $100 \times 100$ | 6.32 | 2.27 | 6.32 | 2.27 |
| | | | $120 \times 120$ | 6.40 | 1.27 | 6.40 | 1.27 |
| | | | $140 \times 140$ | 6.45 | 0.78 | 6.45 | 0.78 |
| 0.5 | 0 | $10^4$ | $100 \times 100$ | 5.78 | – | 5.44 | – |
| | | | $120 \times 120$ | 5.91 | 2.25 | 5.56 | 2.21 |
| | | | $140 \times 140$ | 6.01 | 1.69 | 5.56 | 1.62 |
| | | | $160 \times 160$ | 6.06 | 0.83 | 5.70 | 0.88 |
| 0.5 | 0 | $10^5$ | $120 \times 120$ | 9.57 | – | 9.24 | – |
| | | | $140 \times 140$ | 9.81 | 2.51 | 9.41 | 1.81 |
| | | | $160 \times 160$ | 9.93 | 1.22 | 9.52 | 1.16 |
| | | | $180 \times 180$ | 10.03 | 0.71 | 9.59 | 0.73 |

**Table 2.** Time-step sensitivity analysis for $W/H = 0.5$, $L/H = 0$ and $Ra_\mathrm{H} = 10^5$.

| Mesh Size | $\Delta\tau$ | $Nu_\mathrm{L}$ | % | $Nu_\mathrm{U}$ | % |
|-----------|-----------|------|------|------|------|
| $160 \times 160$ | $10^{-2}$ | 10.24 | – | 9.84 | – |
| | $10^{-3}$ | 10.07 | −1.66 | 9.65 | −1.93 |
| | $10^{-4}$ | 9.93 | −1.39 | 9.52 | −1.35 |
| | $10^{-5}$ | 9.89 | −0.40 | 9.51 | −0.11 |

Finally, with the scope of this study including the validation of both the numerical code and the discretization grid scheme, three different tests have been carried out. In the first test, the average Nusselt numbers obtained for a single vertical plate suspended in free air at several Rayleigh numbers have been compared with the predictions of the Churchill–Chu correlation [29]. In the second test, the average Nusselt numbers obtained for a pair of face-to-face vertical plates have been compared with the predictions of the correlating equation developed by Bar-Cohen and Rosenhow [5]. In the third test, the optimal distance between pairs of face-to-face vertical plates calculated at different Rayleigh numbers have been compared with the numerical results obtained by Olsson [30]. It is apparent that a satisfactory degree of agreement between our numerical results and the literature data was achieved in any validation test carried out; this is shown in Figure 2 and in Tables 3 and 4.

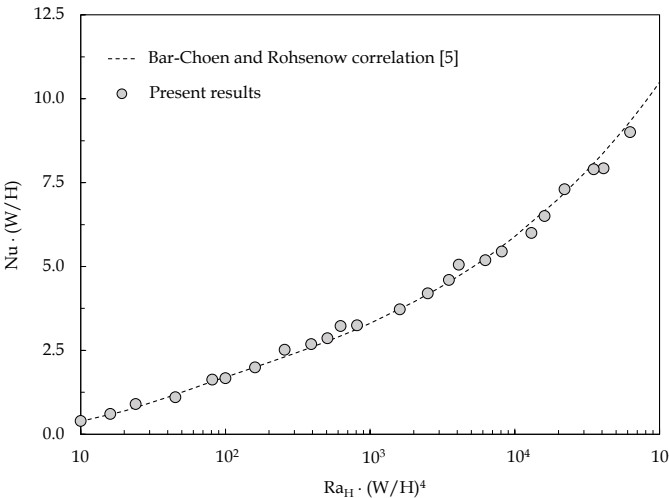

**Figure 2.** Comparison between present numerical data and Bar-Choen and Rohsenow correlation [5] for two vertical parallel plates suspended in free air.

**Table 3.** Comparison between the present numerical data and the predictions of the Churchill-Chu correlation [29] for a heated vertical plate suspended in free air.

| $Ra_{\mathrm{H}}$ | $Nu_{\mathrm{H}}$ **(Present Study)** | $Nu_{\mathrm{H}}$ **[29]** |
|:---:|:---:|:---:|
| $10^3$ | 3.26 | 3.43 |
| $10^4$ | 5.43 | 5.43 |
| $10^5$ | 9.33 | 9.21 |
| $10^6$ | 17.05 | 16.56 |

**Table 4.** Comparison between the numerical data obtained for $(W/H)_{\mathrm{opt}}$ and the values derived by Olsson [30] for $Ra_{\mathrm{H}} = 10^4 - 10^6$.

| $Ra_{\mathrm{H}}$ | $(W/H)_{\mathrm{opt}}$**(Present Study)** | $(W/H)_{\mathrm{opt}}$ **[30]** |
|:---|:---:|:---:|
| $10^6$ | 0.18 | 0.17 |
| $10^5$ | 0.32 | 0.31 |
| $10^4$ | 0.57 | 0.52 |

## 3. Results

Numerical simulations have been carried out for different values of (a) the Rayleigh number $Ra_{\mathrm{H}}$ in the range between $10^4$ and $10^6$; (b) the dimensionless horizontal spacing between the plates $W/H$ in the range between 0.1 and 1; and (c) the dimensionless vertical alignment of the plates $L/H$ in the range between 0 (plates fully staggered) and 1 (plates located face to face). In all the simulations executed, the value of the Prandtl number has been set equal to 0.7, which corresponds to air.

Based on the collection of computed velocity and temperature fields, the main local and overall heat transfer features will be analyzed first. Subsequently, adequate sets of correlations for the average Nusselt numbers will be constructed and discussed.

A selection of typical local results at steady state, displayed in the form of isotherm contour plots, is reported in Figures 3–5 for the following combinations of the independent variables: (a) $Ra_{\mathrm{H}} = 10^5$, $W/H = 0.25$ and $L/H = 1$, 0.75, 0.5, 0.25 and 0; (b) $Ra_{\mathrm{H}} = 10^5$, $L/H = 0.5$ and $W/H = 0.25$, 0.5, 0.75 and 1; and (c) $L/H = 0.5$, $W/H = 0.25$ and $Ra_{\mathrm{H}} = 10^4$, $10^5$, and $10^6$.

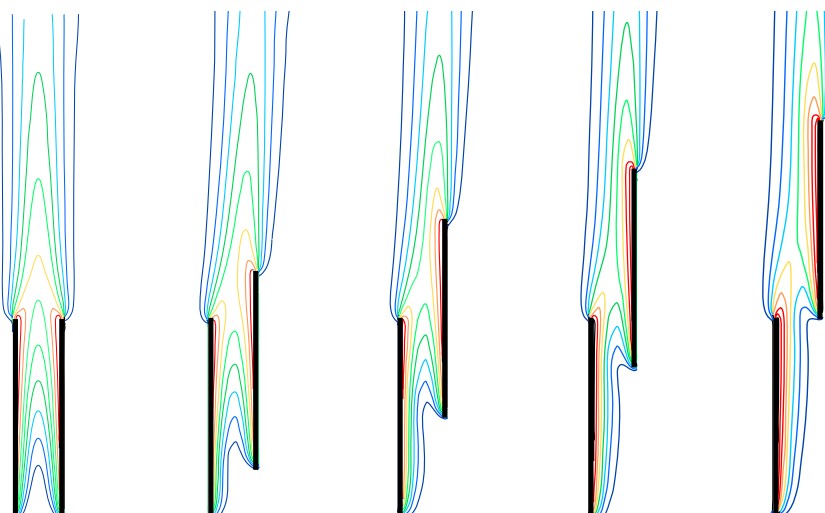

**Figure 3.** Isotherm contour plots for $Ra_{\mathrm{H}} = 10^5$, $W/H = 0.25$ and $L/H = 1, 0.75, 0.5, 0.25, 0$ using a scale from blue $(T = 0)$ to red $(T = 1)$.

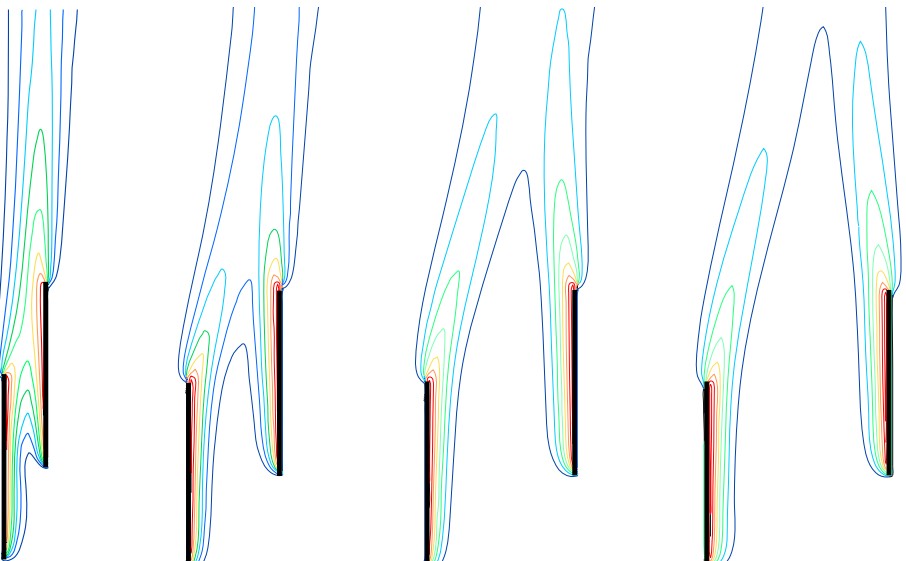

**Figure 4.** Isotherm contour plots for $Ra_{\mathrm{H}} = 10^5$, $L/H = 0.5$ and $W/H = 0.25, 0.5, 0.75, 1$ using a scale from blue $(T = 0)$ to red $(T = 1)$.

It can be observed that when the plates are either located face to face or moderately staggered, the temperature distribution is typical for the channel flow and the thermal performance of both plates is influenced by the well-known chimney effect provided that the separation distance between the plates is neither too small to limit the fluid flow rate due to the marked friction effects and not too large to cause both plates to behave as a single plate. Of course, the chimney effect decreases drastically as the plates are more and more staggered; the effect could vanish when the plates are considerably or fully staggered. In such a case, two distinct boundary layers tend to form along the heated side of each plate. However, while the lower plate tends to actually behave as a single plate, at short separation distances the thermal performance of the upper plate is affected by the concurrence of the two opposite effects which originate from the warm plume spawned by the lower plate. In fact, the hot buoyant flow from the lower plate acts as a forced convection field wherein the upper plate is embedded, which tends to enhance the heat transfer rate at the upper plate surface; on the other hand, the upward-moving warm plume causes a decrease in the temperature difference between the upper plate and the adjacent fluid, which tends to decrease the heat transfer rate at the upper plate

surface. Accordingly, for any vertical alignment configuration, the existence of an optimal dimensionless separation distance between the plates related to the Rayleigh number has to be envisioned, such that the flow rate enhancement due to the chimney effect is large enough and, at the same time, the temperature of the fluid stream rising along the upper plate is close enough to that of the undisturbed fluid reservoir. Of course, such an optimal separation distance also depends on the fact that the maximum heat transfer rate has to be attained for the lower plate or for the upper plate or for the pair of staggered plates.

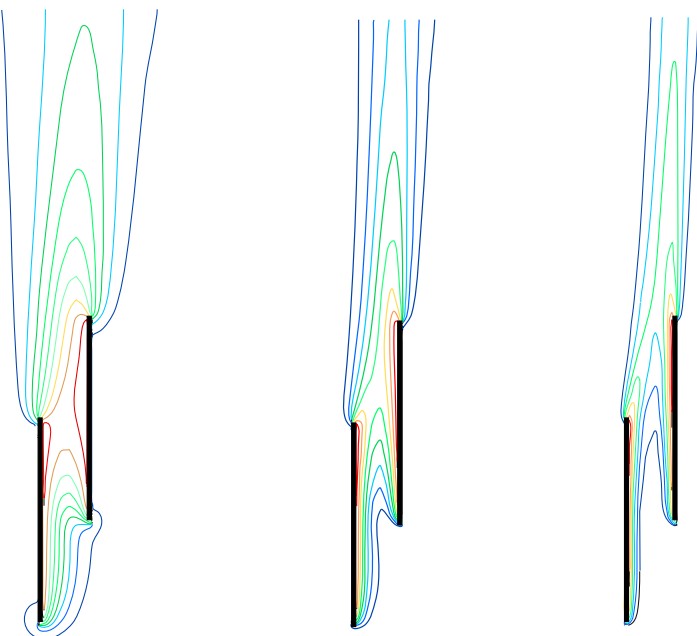

**Figure 5.** Isotherm contour plots for $W/H = 0.25$, $L/H = 0.5$ and $Ra_{\mathrm{H}} = 10^4$, $10^5$, $10^6$ using a scale from blue ($T = 0$) to red ($T = 1$).

This is clearly shown in Figures 6–8 and in Figures 9–11, where the distributions of the average Nusselt numbers $Nu_{\mathrm{L}}$ and $Nu_{\mathrm{U}}$ for the lower and upper plates and the distributions of the average Nusselt number $Nu$ for the whole system, respectively, are depicted for the same combinations of independent variables used earlier to delineate the local solutions. In Figures 6–8, the Nusselt numbers for a single vertical plate at the same Rayleigh number listed in Table 3 are also reported for comparison purposes using dashed lines.

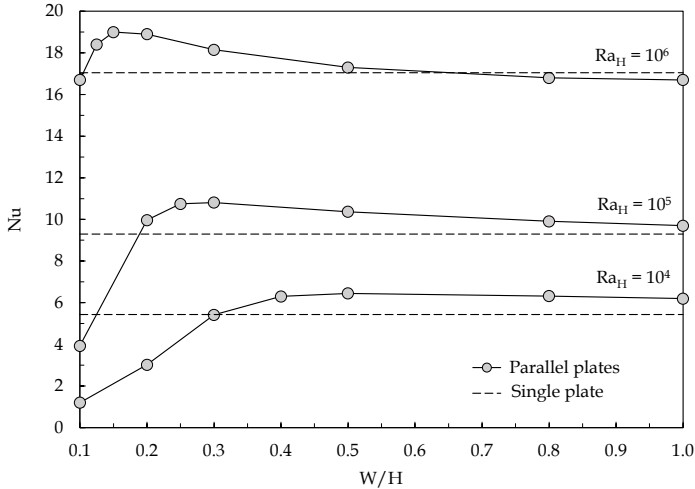

**Figure 6.** $Nu$ vs. $W/H$ for $L/H = 1$ (face-to-face plates) using $Ra_{\mathrm{H}}$ as the parameter.

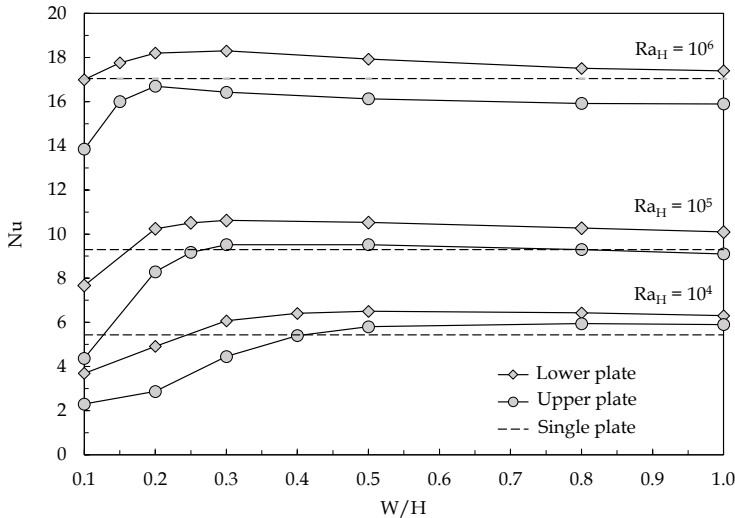

**Figure 7.** *Nu* vs. *W/H* for *L/H* = 0.5 using *Ra*$_\text{H}$ as the parameter.

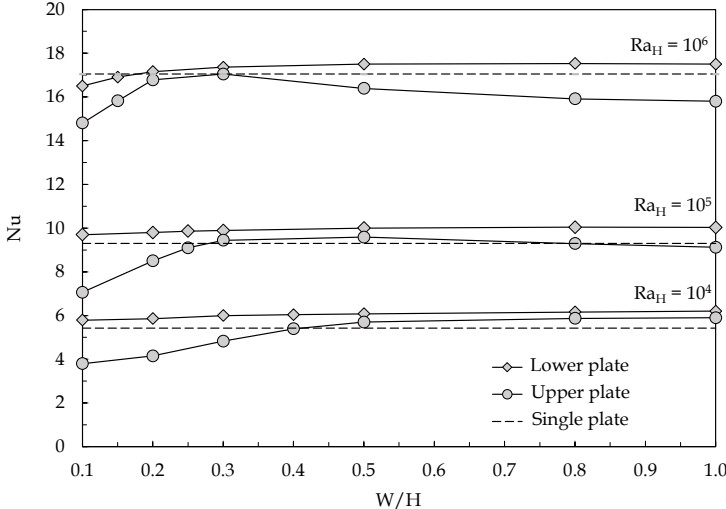

**Figure 8.** *Nu* vs. *W/H* for *L/H* = 0 (fully staggered plates) using *Ra*$_\text{H}$ as the parameter.

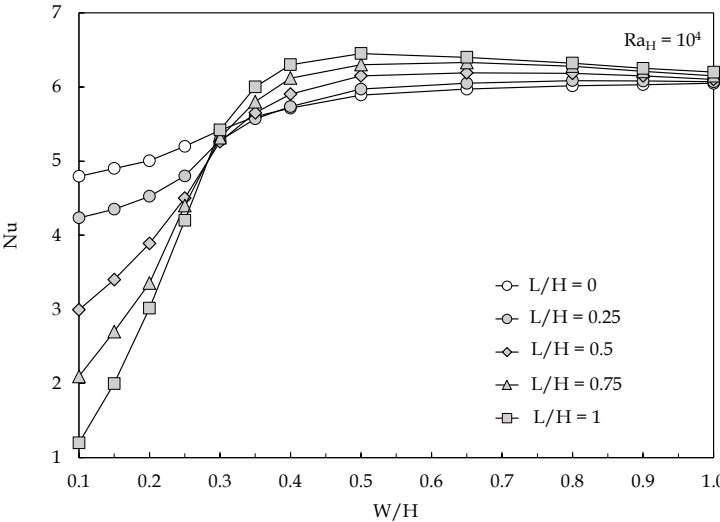

**Figure 9.** *Nu* vs. *W/H* for *Ra*$_\text{H}$ = 10$^4$ using *L/H* as the parameter.

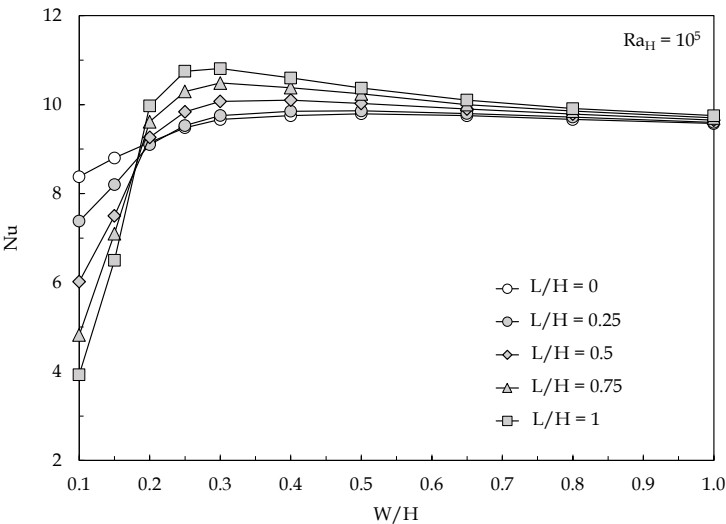

**Figure 10.** $Nu$ vs. $W/H$ for $Ra_H = 10^5$ using $L/H$ as the parameter.

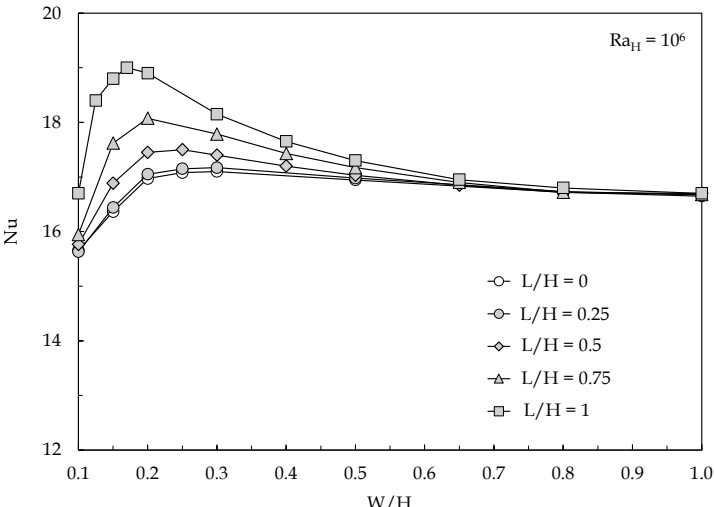

**Figure 11.** $Nu$ vs. $W/H$ for $Ra_H = 10^6$ using $L/H$ as the parameter.

In focusing the attention on the thermal performance of the whole system, it can be observed that, at large plate spacings, the heat transfer rate is almost independent of the plate staggering as each plate tends to behave as a single plate, whereas, at intermediate separation distances, the average Nusselt number increases as the plate staggering is decreased, which is due to the aiding contribution of the chimney effect. Conversely, at close spacing between the plates, the average Nusselt number enhances as the plate staggering is increased due to the larger portion of each plate being exposed to the undisturbed fluid reservoir. Such a behavior inversion occurs at a plate spacing $(W/H)_\text{inv}$, which decreases with increasing the Rayleigh number.

The distributions of the optimal dimensionless separation distance between the plates $(W/H)_\text{opt}$ for the maximum heat transfer rate from the pair of staggered plates, which are plotted against the Rayleigh number $Ra_H$, are displayed in Figure 12 using the dimensionless vertical alignment $L/H$ as a parameter. It can be observed that $(W/H)_\text{opt}$ decreases both as $Ra_H$ increases due to the decrease of the boundary layer thickness and $L/H$ decreases due to the reduced interaction occurring between the bottom portion of the boundary layer adjacent to the upper plate and the top portion of the boundary layer adjacent to the lower plate.

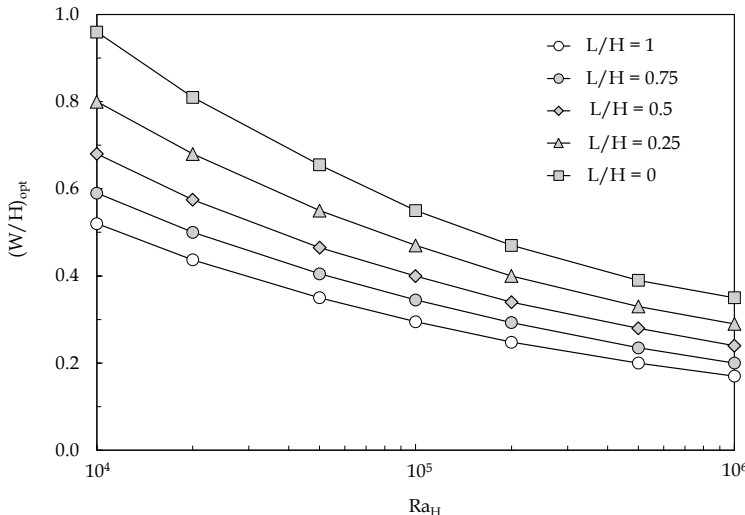

**Figure 12.** $(W/H)_{opt}$ vs. $Ra_\mathrm{H}$ using $L/H$ as the parameter.

The whole set of numerical results obtained for the optimal distance $(W/H)_\mathrm{opt}$ and the corresponding peak-value of the average Nusselt number, denoted as $Nu_\mathrm{opt}$, can be correlated by using the following pair of dimensionless correlating equations obtained using a multiple regression method:

$$\left(\frac{W}{L}\right)_\mathrm{opt} = 8 \cdot Ra_\mathrm{H}^{-0.23}\left(1 + \frac{L}{H}\right)^{-0.9} \tag{8}$$

$$Nu_\mathrm{opt} = 0.69 \cdot Ra_\mathrm{H}^{0.23}\left(1 + \frac{L}{H}\right)^{0.16} \tag{9}$$

where the former has a 2.4% standard deviation of error and the latter with a 1.9% standard deviation of error, as shown in Figures 13 and 14, respectively.

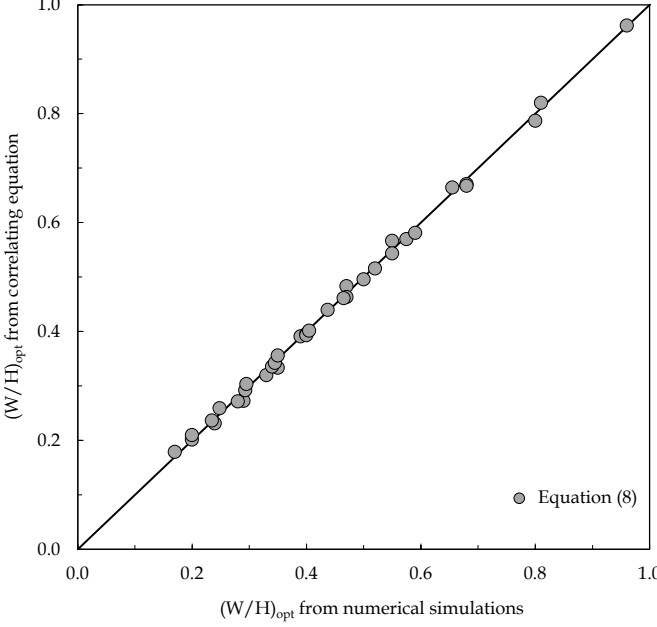

**Figure 13.** Comparison between Equation (8) and the numerical data obtained for $(W/H)_\mathrm{opt}$.

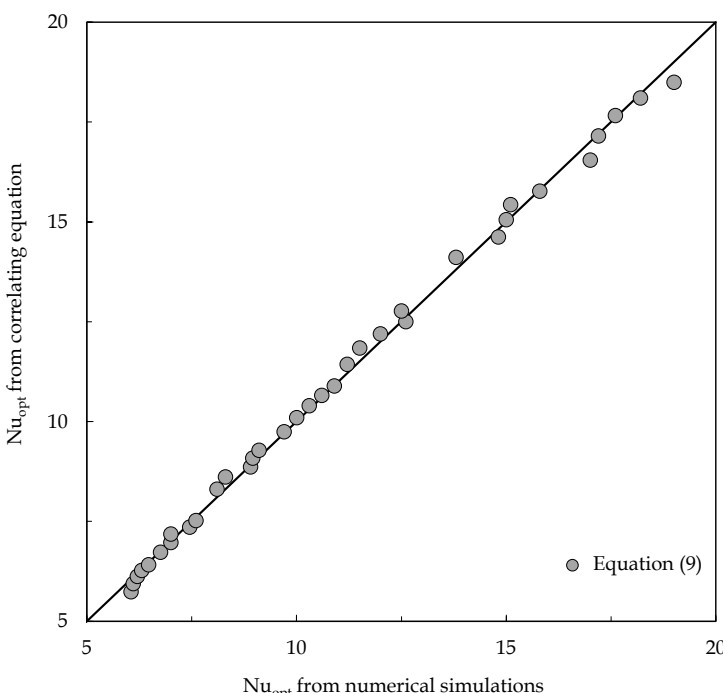

**Figure 14.** Comparison between Equation (9) and the numerical data obtained for $Nu_\text{opt}$.

In addition, a set of heat transfer dimensionless correlations is developed for predicting the average Nusselt numbers for the whole system and for each plate and for separation distances between the plates in the range between $(W/H)_\text{opt} < W/H \leq 1$:

$$Nu = 0.74 \cdot Ra_\text{H}^{0.223} \left(\frac{W}{H}\right)^{-0.053} \left(1 + \frac{L}{H}\right)^{0.05} \tag{10}$$

$$Nu_\text{L} = 0.76 \cdot Ra_\text{H}^{0.225} \left(\frac{W}{H}\right)^{-0.052} \left(1 + \frac{L}{H}\right)^{0.02} \tag{11}$$

$$Nu_\text{U} = 0.72 \cdot Ra_\text{H}^{0.222} \left(\frac{W}{H}\right)^{-0.055} \left(1 + \frac{L}{H}\right)^{0.09} \tag{12}$$

and for separation distances in the range between $(W/H)_\text{inv} < W/H < (W/H)_\text{opt}$:

$$Nu = 0.53 \cdot Ra_\text{H}^{0.26} \left(\frac{W}{H}\right)^{0.107} \left(1 + \frac{L}{H}\right)^{0.18} \tag{13}$$

$$Nu_\text{L} = 0.63 \cdot Ra_\text{H}^{0.246} \left(\frac{W}{H}\right)^{0.07} \left(1 + \frac{L}{H}\right)^{0.16} \tag{14}$$

$$Nu_\text{U} = 0.445 \cdot Ra_\text{H}^{0.276} \left(\frac{W}{H}\right)^{0.147} \left(1 + \frac{L}{H}\right)^{0.2} \tag{15}$$

where, according to our data, $(W/H)_\text{inv}$ can be easily evaluated as follows.

$$(W/H)_\text{inv} = 2.24 \cdot Ra_\text{H}^{-0.22} \tag{16}$$

The standard deviations of the errors are 2.3%, 2.75% and 3.25% for Equations (10)–(12) and 3.7%, 3.2% and 4.9% for Equations (13)–(15), respectively, as displayed in Figure 15.

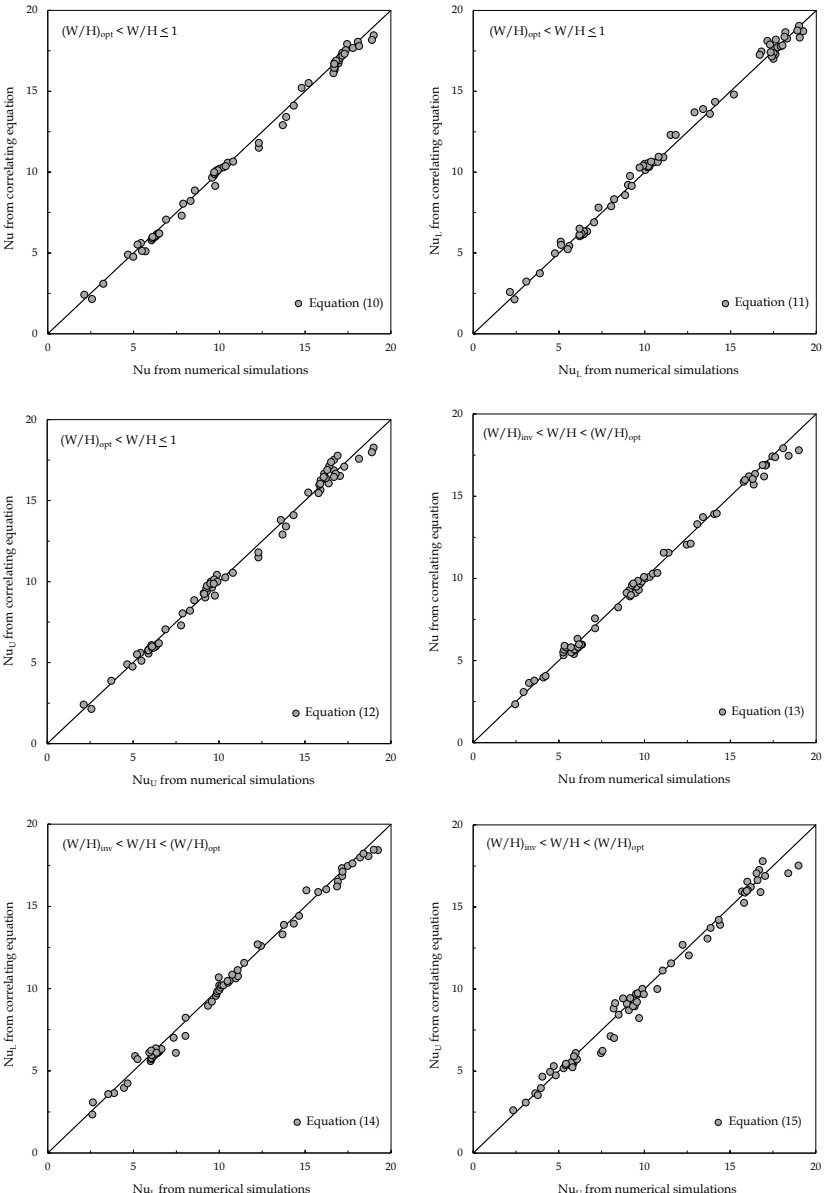

**Figure 15.** Comparison between Equations (10)–(15) and the numerical data obtained for the Nusselt numbers of the whole system, the lower plate and the upper plate for $\frac{W}{H} \neq \left(\frac{W}{H}\right)_{\text{opt}}$.

## 4. Conclusions

Buoyancy-induced convection from a pair of thin staggered heated vertical plates suspended in free air has been studied numerically using a control-volume formulation of the finite-difference method based on the SIMPLE-C algorithm. The investigation has been performed by using the Rayleigh number based on the plate length, as well as the horizontal separation distance between the plates and their vertical alignment as independent variables.

The main results obtained in the present study may be summarized as follows:

(a) The existence of an optimal horizontal spacing for maximum heat transfer rate has been found at any investigated Rayleigh number and vertical alignment;
(b) The optimal plate spacing decreases when increasing both $Ra_H$ and $L/H$;
(c) For high values of $W/H$, the Nusselt number is independent of $L/H$ since each plate tends to behave as a single plate;

(d) For intermediate values of $W/H$, the Nusselt number increases as $L/H$ is increased due to the contribution of the chimney effect;

(e) For low values of $W/H$, the Nusselt number increases as $L/H$ is decreased due to the larger portion of each plate exposed to free air.

**Author Contributions:** Conceptualization, A.Q. and M.C. (Massimo Corcione); methodology, A.Q., M.C. (Massimo Corcione) and I.P.; software, A.Q. and V.A.S.; validation, A.Q., M.C. (Marta Cianfrini) and V.A.S.; data curation, A.Q. and V.A.S.; writing—original draft preparation, A.Q. and M.C. (Massimo Corcione); writing—review and editing, A.Q., M.C. (Marta Cianfrini), M.C. (Massimo Corcione), I.P. and V.A.S. All authors have read and agreed to the published version of the manuscript.

**Funding:** This research received no external funding.

**Conflicts of Interest:** The authors declare no conflict of interest.

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
