# Peer review of "Dimensionless Correlations for Natural Convection Heat Transfer from a Pair of Vertical Staggered Plates Suspended in Free Air"

_applsci, doi:10.3390/app11146511_

Round 1
Reviewer 1 Report
The reviewer likes the correlations derived as a result of this investigation. The study would be suitable for publication once the following suggestions are implemented. Therefore, the reviewer recommends a major revision of the manuscript before publishing the article in the Applied Sciences.
- Equation (2) is missing viscosity (or Reynolds number in the normalized case) in the viscous term (the second term on the right-hand side of the equation). The reviewer suggests authors carefully review all of their programs for the accuracy of these equations. If this equation is not accurate the entire database is not valid for the study.
- Equations (2), and (3) also include the transient term. Do you consider any transient flow effects in this study? If it is not the reviewer suggests removing the transient term from the equations.
- The authors present that the analysis is conducted only for laminar flows. The reviewer strongly recommends including plots that illustrate the velocity profiles across two plates for different configurations. These plots would be beneficial to make sure that the flow regime is laminar or turbulent. Because the reviewer presumes that the flow over the upper plate (2nd plate) could possibly be turbulent particularly when the plates are sufficiently staggered. Because the already developed boundary layer over the first plate significantly affects the flow regime over the second plate. Therefore, it is better to investigate the flow over the plates prior to the heat transfer analysis.
- Contour values of any of the figures are not shown. It would be helpful for the reader if the contour values are included in the figures.
Author Response
First of all, the authors wish to thank the Reviewer for her/his valuable comments. The manuscript has been properly modified to address the reviewer’s requests. A point-by-point response is provided in the following section.
1) Due to the buoyancy driven motion, in the normalized momentum equation (2) the Rayleigh number, Ra = Gr x Pr, is the relevant dimensionless parameter rather than the Reynolds number, which is typical of a forced convection flow.
2) The study is conducted using a transient formulation of the problem up to reaching a steady-state solution, which ensures to achieve a more accurate convergence than the direct steady state formulation.
3) Due to the moderate Rayleigh numbers considered in the present study, the analysis is conducted under the assumption of laminar flow. On the other hand, also with reference to the highest investigated Rayleigh number of 106, for completely staggered plates the maximum “overall” Rayleigh number based on twice the plate length would be 8x106, which safely corresponds to laminar flow, taking into account that transition on a vertical plate typically occurs around Ra=109.
4) The contours of Figs. 3-5 represent the dimensionless temperatures 0 and 1 using a step of 0.1. For better clarity, these figures have been enlarged and represented adopting colors, using a scale from dark blue (for T=0) to bright red (for T=1), as specified in the new captions.
Reviewer 2 Report
I recommend the manuscript for publication. But I advise you to rework Figures 3-5. They contain useful data, but it is completely incomprehensible where which isotherm is located. You can add scales or colors, or mark values side by side, or use different types of lines, etc.
Suggestions for further research.
It is clear why this geometry was chosen. But it would be interesting to make a comparison with some kind of experiment. You can use the data on the flow rate at some distance from the plates. Or compare density distributions with shadow patterns. Such experiments were actively carried out in the middle of the last century. You can see "An Album of Fluid Motion" by Milton Van Dyke and similar sources.
Author Response
First of all, the authors wish to thank the Reviewer for her/his valuable comments. The manuscript has been properly modified to address the reviewer’s requests. A point-by-point response is provided in the following section.
1) The contours of Figs. 3-5 represent the dimensionless temperatures 0 and 1 using a step of 0.1. For better clarity, these figures have been enlarged and represented adopting colors, using a scale from dark blue (for T=0) to bright red (for T=1), as specified in the new captions.
Round 2
Reviewer 1 Report
Accepted for publication.